# Graphene Optical Biosensors

**DOI:** 10.3390/ijms20102461

**Published:** 2019-05-18

**Authors:** Zongwen Li, Wenfei Zhang, Fei Xing

**Affiliations:** School of Physics and Optoelectronic Engineering, Shandong University of Technology, Zibo 255049, China; zwli@stumail.sdut.edu.cn

**Keywords:** Graphene, optical, surface wave, biosensors

## Abstract

Graphene shows great potential in biosensing owing to its extraordinary optical, electrical and physical properties. In particular, graphene possesses unique optical properties, such as broadband and tunable absorption, and strong polarization-dependent effects. This lays a foundation for building graphene-based optical sensors. This paper selectively reviews recent advances in graphene-based optical sensors and biosensors. Graphene-based optical biosensors can be used for single cell detection, cell line, and anticancer drug detection, protein and antigen–antibody detection. These new high-performance graphene-based optical sensors are able to detect surface structural changes and biomolecular interactions. In all these cases, the optical biosensors perform well with ultra-fast detection, high sensitivities, unmarked, and are able to respond in real time. The future of the field of graphene applications is also discussed.

## 1. Introduction

Graphene, the thinnest material known [1,2,3,4], has attracted attention of scientists worldwide in recent years [5,6]. Two scientists at the University of Manchester, Andre Geim and Konstantin Novoselov, won the 2010 Nobel Prize in Physics for their pioneering experiments on graphene in the field of two-dimensional materials [7,8,9,10]. Graphene is recognized as a revolutionary material due to its remarkable optical, electronic properties [1,11,12,13]. These excellent properties of graphene are derived from its electronic structure and lattice structure. Graphene has a two-dimensional (2D) hexagonal packed structure of sp^2^ hybridization with remarkable electronic properties [14,15,16]. Figure 1 shows the electron band structure and the Dirac point of graphene. The Fermi surface of graphene is in the intersection of the completely filled valence band and the empty conduction band, as well as in the middle of the p band, where electrons are the valence electrons [16]. These valence electrons in monolayer graphene are massless Dirac fermions, and their mobility is 1/300 of the speed of light [17]. Graphene shows a conductivity of up to 10^6^ s/m and resistivity of only about 10^−6^ Ω⋅cm, which is lower than copper or silver and is the material with the lowest resistivity in the world [18,19,20]. Therefore, graphene also has the same electrical conductivity as copper [21,22,23].

Due to the special electron band structure, graphene shows high carrier mobility and zero bandgap characteristics. Thus, the excellent optical properties of graphene, such as broadband absorption, saturation absorption, and fluorescence, gradually attract the attention of scientists [6,24,25,26,27,28]. These optical properties of graphene mainly result from its special lattice structure and electronic structure. Although the thickness of monolayer graphene is 3.35 Å, its absorbance is high, and due to the linear distribution of the Dirac electrons, each layer of graphene absorbs 2.3% of the light from the visible to the terahertz broad band. Absorption of graphene has been experimentally observed to have a universal value ≈πα (2.3%) for light in the visible spectral range, depending on the fine structure constant α, but not on the properties of the material [6,29]. This value of optical absorption indicates that graphene shows an ultra-high transparency in the field of materials [6]. Due to the ultra-fast kinetics of Dirac electrons and the presence of Pauli blocking in the tapered band structure, it imparts excellent nonlinear optical properties to graphene. In addition, Dirac electrons in graphene exhibit a linear energy-momentum dispersion relationship near the Dirac point, which allows graphene to have a resonant optical response to photons at any frequency in the ultra-broadband spectral range of the ultraviolet-visible-infrared region [24]. Because of its strong broadband absorption, graphene exhibits different reflectance for transverse electric (TE) and transverse magnetic (TM) modes under total internal reflection, which is sensitive to the refractive index (RI) of the media in contact with the surface [30,31,32,33]. For graphite sheets, when the energy of electrons and holes is close to the Dirac point, pairs of carriers can form interband transitions by collision [34]. Therefore, monolayer graphene can provide the highest saturation absorption value. Graphene can produce fluorescence by a special method to satisfy a certain band gap [35].

There are two main types of interactions between light and graphene from the energy band transition: Interband transitions and in-band transitions. In the far-infrared and THz spectral regions, the electronic response is mainly in-band transition (free carrier response), which can be well described by the Drude model. The graphene electron response in this band is similar to the free electron response in the metal, which can excite the surface. In the near-infrared and visible-light bands, the photoresponse is mainly interband transition. The absorption of light appears as a general absorption independent of wavelength, and the absorption coefficient is determined by the fine structure constant. In this case, due to the Pauli blocking principle, the light absorption of graphene can be regulated by adjusting the position of the Fermi surface. In the ultraviolet region, the interband transition is close to the saddle point, at which point the light absorption exceeds the universal absorption value and has an exciton effect. Graphene shows high optical absorption as a monoatomic layer material, but as a material, 2.3% optical absorption is still very low. This suggests that the interaction of light with graphene is still weak, and many of the optical properties of graphene are limited by this weaker optical absorption. Therefore, it is a prominent problem of graphene optics to find ways to enhance the interaction between light and graphene.

Moreover, graphene also exhibits excellent biological [36,37,38,39], chemical [1,15,40,41,42], thermal [43,44,45], mechanical [46,47,48,49], and magnetic [50,51,52] properties, it has shown great potential applications in many fields, such as sensing [53,54], optical inspection [55,56], electronics [57,58], biomedicine [38,59], etc., and has attracted strong scientific interest, which has set off a research boom around the world.

These excellent properties of graphene are used in many different types of sensing applications, including optical sensors, electric field sensors, electrochemical sensors, and gas sensors. First of all, the excellent electrical properties of graphene have aroused wide interest of scientists. Graphene has good conductivity, high current density of sustainability, high carrier mobility and relativistic carrier property [60]. Thus, the graphene-based field effect transistor (FET) biosensor was developed to sensitively detect the current or resistive characteristics of the cell membrane, and it can show rapid response speed [61,62]. In 2011, Ang developed a graphene transistor array (shown in Figure 2a) integrated with microfluidic flow cytometry for sensing of malaria-infected red blood cells [63]. These cells induced highly sensitive capacitively coupled changes in the conductivity of graphene, from which detailed information on the condition can be obtained. The left image of Figure 2b shows the conductivity of the two cells (trophozoite-Plasmodium falciparum-infected erythrocyte (PE) or schizont-PE) as they pass through the functionalized graphene surface over time. As shown in Figure 2c, at the charge neutrality point (CNP), different conductivities represent the ability to distinguish between the developmental stages of trophozoites and schizonts [61]. Carbon nanotube is the most widely used material in the field of electrochemical analysis and electrocatalysis [64]. However, the graphene-based electrode has more advantages in electrocatalytic activity and macroscopic conductivity than carbon nanotubes. Thus graphene has a good chance to play a key role in the field of electrochemistry [65,66]. The graphene-based electrode not only detects infected cells but also shows good electrocatalytic activity for drugs. Kang proposed an electrochemical sensor based on the electrocatalytic activity of functionalized graphene, which is useful for sensitive detection of acetaminophen [67]. In the experiment, the relationship between the concentration of NH_3_·H_2_O–NH_4_Cl (0.1 M) buffer solution (c) and the oxidation peak current (I_pa_) of acetaminophen was studied by square wave voltammetry (SWV). The I_pa_ was linearly related to the paracetamol concentration over the range of 0.1–20 μM. The results showed that the graphene-modified glassy carbon electrode (GCE) could exhibit good reproducibility when detecting paracetamol. Cyclic voltammetry (CV) was used to compare the interference of dopamine acid (DA) and ascorbic acid (AA) on bare GCE and graphene-modified GCE, respectively. These results indicate that the developed sensor exhibits an excellent electrocatalytic activity towards the reducing and oxidation of paracetamol. Moreover, it has an excellent detection limit of paracetamol with a detection limit of 3.2 × 10^−8^ M, which has high sensitivity and selectivity for detecting paracetamol in medical applications and commercial fields. All in all, the electrode modified with graphene had good electrocatalytic activity to paracetamol.

In short, in the field of electrochemistry, graphene has the opportunity to show its talents. The application of graphene on electrochemical sensors has the advantages of high sensitivity, fast response time, fast electron transfer, easy to fix proteins and maintain their activity. However, there are some shortcomings in graphene-based electrical sensors. Since transistor-based graphene sensors only measure current changes on the surface of graphene, electrical sensing is seriously restrained compared to the time and spatial resolution of optical sensing [30]. In addition, electrical measurements usually damage living cells, which could have an impact on test results. Therefore, studying the optical properties of graphene and developing graphene-based optical sensors can solve these problems well. The difference between the graphene-based electrical sensor and the graphene-based optical sensor is shown in Table 1.

Graphene shows unique and ideal optical properties, such as broadband and tunable absorption [12,69], and polarization-dependent effects [13,70]. Therefore, the optical properties of graphene establish the foundation for building graphene-based optical devices, and graphene-based optical sensors have been developed successively [71]. At present, graphene-based optical sensors mainly include surface plasmon resonance (SPR) sensors, graphene optical fiber SPR sensors and graphene spatial light sensors. Among them, the most widely used optical sensor is the sensor based on SPR [68], which is highly sensitive, unmarked, and able to respond in real time. Moreover, SPR biosensor is characterized by real-time monitoring, convenient and fast detection, high sensitivity and high-quality analysis data, tracking and monitoring of ligand stability, ensuring reaction balance and wide applications [72]. For optical sensors, it is important to measure the rate of change of the refractive index in various applications, which can be applied to biosensors [73,74,75]. Graphene-based optical biosensors can be used for single cell detection [32,76,77], cell line and anticancer drug detection [78,79,80], protein and antigen–antibody detection [81,82,83]. These new high-performance graphene optical sensors will be able to detect changes in the surface structure of graphene and its interaction with biomolecules [68,84].

## 2. Graphene Optical Sensor Based on Surface Plasmon Resonance

Graphene has excellent high-energy transfer efficiency, large surface area, and biocompatibility [32], and can be applied to optical sensors to detect different samples such as cells, proteins, and small molecules [85]. Wang, who fabricated the new sensor, said: “this is the first time that pure graphene has been used to create a high-sensitivity sensor with a wide range of applications.” They confirmed that using graphene alone can create a cheap and flexible photosensitive sensor. Wang also claimed that the key to the new sensor is that the “trapped light” nanostructures can capture light from electronic particles for longer periods of time than traditional sensors [60]. These sensors benefit from the excellent optical properties of graphene materials, which not only provide researchers with very useful information but have also achieved great success in recent years [84,86].

### 2.1. Graphene Surface Plasmon Resonance Biosensors

Surface plasmon is quantum of plasmonic oscillation generated by the interaction of freely vibrating electrons and photons on a metal surface, which was observed in Wood’s experiment in 1902 [87]. SPR is the phenomenon that, when the polarized light of parallel surface illuminates at the interface between glass and metal film with a certain angle to attenuate and fully reflect, the incident light is coupled into the surface plasmon, and the evanescent wave penetrating into the metal film will excite free electrons in the metal to produce surface plasmon, as shown in Figure 3a. When the incidence angle or wavelength reaches a certain value, the frequency and wave number of surface plasmon are equal to that of the evanescent wave, and the two will produce resonance. In this situation, the incident light is coupled into the surface plasmon, and the light energy is seriously absorbed, resulting in a sharp decline in the reflected light energy, and a resonance peak (the lowest reflection intensity) appears on the reflection spectrum [88,89,90]. The wave vector of the surface plasmon (*k_sp_*) propagating at the metal-dielectric interface is described by the following equation [89],
(1)$$k_{sp}=\frac{\omega_{sp}}{c}\sqrt{\frac{\varepsilon_{m}\varepsilon_{s}}{\varepsilon_{m}+\varepsilon_{s}}}$$
where *ω_sp_* is the angular frequency of the surface plasmon, and *ε_m_* and *ε_s_* are the dielectric constants of the metal film and the dielectric medium, respectively.

SPR biosensing was first applied in 1983, and since then, biospecific interactions have also begun to be detected [88]. Since the development of the first biosensor based on SPR, the use of this technique has increased steadily. In the traditional SPR biosensor structure, a thin metal film is adhered to the hypotenuse of the prism to separate the sensor media from the prism. Metal membranes are usually made from gold [91,92] or silver [93] because the plasma frequencies of both metals are in visible parts of the electromagnetic spectrum [68]. However, there are some unavoidable defects in the sensors based on gold and silver materials. For example, silver is oxidized when exposed to air, performance is reduced and life is shortened. Biomolecules show poor adsorption on gold, limiting sensitivity, The spectral broadening caused by inter-band transitions also obstructs the performance of the sensor [68,94]. In addition, the weak biocompatibility of the gold and silver membranes may cause cell rejection and produce false signals [95]. In view of the defects of sensors based on gold and silver film, graphene-based sensor has been developed. The deposition of graphene on the metal layer enhances the electric field and biological sensing ability, thus advancing the performance of the sensor. Moreover, graphene also helps to adsorb biomolecules better, as a result of π-π stacking, which increases the system’s affinity for these molecules [94].

In 2010, Wu presented an SPR-based graphene biosensor by coating graphene over a gold film [68]. Compared with traditional SPR biosensor devices, the use of graphene as a biomolecular recognition element (BRE) coated on the gold surface not only improves the adsorption efficiency of biomolecules, but also detects the change in refractive index near the surface of the sensor by using an attenuated total reflection (ATR) method [96]. In the case of simulation and actual measurement, as the thickness of graphene increases, the transmission decreases, and the simulation results are basically consistent with the measured spectra, as shown in Figure 3b. Specifically, for each additional graphene layer, the light transmission is absorbed by an additional 2.3% [68]. The rate of change of the sensor output P reflects the sensitivity of the biosensor. For example, the change in the SPR angle relative to the amount of the measured object. For each SPR curve, the angle of incidence corresponding to the ATR minimum is called the SPR angle under resonance conditions. In addition, due to the adsorption of biomolecules, the SPR curve moved to a larger SPR angle, as shown in Figure 3c. Comparing the common SPR biosensor with the single-layer graphene biosensor, there are SPR angle offsets of S_RI_^0^ = 52 and S_RI_^1^ = 53.2 for ΔP = 0.26, respectively. Besides, adding more graphene layers will widen the SPR curve and make it difficult to measure SPR. Furthermore, there is a linear relationship between the ΔS_RI_^L^/S_RI_^0^ and the number of graphene layers L, as shown in Figure 3d. As the number of graphene layers increases, ΔS_RI_^L^/S_RI_^0^ will increase [68], where the sensitivity increment of the SPR biosensor was ∆S_RI_^L^ = S_RI_^L^ − S_RI_^0^. Due to the optical properties of graphene, it has been found that coating more graphene layers can provide increased sensitivity [97,98]. In summary, the optical properties of graphene and the adsorption of biomolecules on graphene are key factors for improving sensor sensitivity.

In addition, optical focusing limits the sensitivity, resolution, dynamic range, and other functions of the spectral-based SPR instrument when measuring wavelength or angle ranges and produces significant defects in the real-time performance of the mentioned biosensor. There is an inverse relationship between the response speed of the SPR biosensor and its sensitivity. The intensity type SPR sensor performs a fast response speed but low sensitivity. While angle or phase type SPR sensors exhibit higher sensitivity at the expense of lower response speed [99]. Although angle, phase, and spectrum modulated SPR sensors show very good performance (high sensitivity and resolution) with the help of sophisticated and expensive operating equipment, which means the high inspection cost in practical applications. Amplitude-modulated SPR sensors can implement fast, real-time monitoring, and imaging functions, but its performance is usually worse than the SPR sensors mentioned above [87,100]. Up to now, among the different sensors, phase SPR imaging [101,102] is the most sensitive technique.

### 2.2. Graphene Optical Fiber Surface Plasmon Resonance Sensors

Fiber optic sensor is an important sensor, and its basic working principle is similar to other sensors. The signal light emitted from the light source enters the sensing area through the optical fiber, where the parameters change, such as the concentration and refractive index of the objects pending measuring, and then modulates the optical properties of the signal light (such as the intensity, wavelength, frequency, phase, polarization state, etc.). The signal light enters the optical detector (such as the optical fiber spectrometer) through optical fiber and converts it into a readable signal to analyze the parameters of the objects pending measuring. In fiber optic sensors, light is used as a carrier for sensitive information, and optical fibers are used as a transmission medium for sensitive information [103]. Fiber optic sensor shows both the characteristics of optical detection, and its advantages mainly include good electrical insulation performance, strong anti-electromagnetic interference ability, corrosion resistance, explosion proof, non-invasive, high sensitivity, flexible optical path, easy to realize remote monitoring of measured signals, easy to connect with computers and so on [104,105].

Graphene is combined with fiber to obtain a graphene-based optical fiber sensor due to its excellent performance [103,104,105], and it has developed rapidly. At present, researches on the combination of graphene and fiber mainly include the fiber mode-locking technology of graphene as a saturable absorber [106], the fiber SPR sensor based on graphene [94] and the fiber evanescent wave sensor based on graphene [107], etc.

In 2013, Kim et al. firstly used chemical vapor deposition (CVD) to transfer graphene on the d-type region of plastic optical fiber, replacing the traditional metal film layer, to build a graphene fiber SPR sensor, and executed corresponding detection of double-stranded DNA and protein-streptavidin as target molecules. The graphene SPR sensor adsorbs the target molecule, and the resonance summit exhibits a characteristic traverse with the change of the target molecule concentration, which effectively proves that graphene can replace the metal film to realize the graphene fiber SPR sensor [104]. Then, Patnaik et al. combined graphene and indium tin oxide (ITO) coated d-type fibers to design an SPR sensor that could be tuned at near-infrared wavelengths and ensured affinity for biomolecules [94]. A cross-sectional view of the sensor is shown in Figure 4a. The sample droplet is on the top of the coating, and the ITO film of the graphene layer is uniformly coated on the surface of the cladding. When the finite element method (FEM) is used, a small portion of the entire structure is selected for ease of calculation. It can be obtained through experimental calculations that the wavelength sensitivity is 4030 and 5700 nm/refractive index units (RIU) at the refractive index of 1.330 and 1.345, respectively, and there is a clear upward trend. Moreover, the sensor can measure the thickness of the bio-layer on the surface of the graphene to evaluate its performance. For example, the resonance wavelength increases linearly with the increase of the thickness of the bio-layer and the shift in the loss spectrum corresponding to the core mode obtained by changing the thickness of the bio-layer. For the defects of the previously proposed sensors, they optimized different parameters and proposed a sensor with a wavelength sensitivity of 5700 nm/RIU and a larger resolution of 1.754 × 10^−5^ RIU. The sensor can not only use the finite element method to obtain the maximum phase matching between modes, but also to detect the thickness of the biological layer.

In 2014, Qiu and Jiang designed a novel graphene tapered fiber sensor for glucose detection by growing a single-layer graphene film on copper foil by CVD and then transferring the single-layer graphene film to the tapered region of the fiber. The tapered fiber (Figure 4b) shows the structure of the sensing area, which includes the fiber core (RI = n_1_), graphene layer (RI = n_2_), and glucose solution (RI = n_3_). Among them, the graphene layer wraps the sensing area. In a fiber of uniform diameter, the evanescent field cannot interact with the surrounding environment of the fiber. However, for tapered fibers, the evanescent field can interact with the surrounding environment. In this experiment, the concentration of the glucose solution is increased alone, and the RI of the glucose solution is also increased. At the same time, this also leads to an increase in penetration depth and large consumption of energy, and the output power is lowered. In addition, it can be seen that the change in the concentration of the glucose solution affects the intensity of the output light. As the concentration of the glucose solution increases, the intensity of the output light gradually decreases. When the glucose solution concentration is increased from 1% to 40%, the output light intensity is also reduced from 29,500 to 27,500. It can also be seen that at each glucose concentration, the output light intensity exhibits a reasonable linear relationship with it. This work could provide a unique strategy for tapered fiber optic sensors with high reliability and repeatability. Then in 2015, Jiang et al. dipped graphene oxide into the u-shaped zone of quartz fiber and prepared u-shaped fiber sensor with evanescent wave absorption response for the detection of alcohol solution. The results show that the sensor with rotation-coated graphene oxide has good accuracy, repeatability, and sensitivity, and the response time and recovery time are within 5 s. The actual wine samples are tested, and the findings are consistent with the alcohol experimental data, laying a foundation for the practicability of the sensor [108].

## 3. Polarization Absorption Enhanced Biosensors

In various photonic and optoelectronic devices, the interaction of graphene with light is also different, so the optical devices of graphene need to work under different structures. Among them, the sensor requires a strong interaction between graphene and light. Using a prismatic TIR structure, the interaction of graphene with light has the characteristic of polarization absorption. Under this structure, graphene shows the characteristics of polarization absorption and broadband absorption enhancement. The polarization absorption enhancement of graphene can be achieved by optimizing the refractive index around the graphene medium or by using a multilayer structure. Based on the polarization absorption effect of graphene under TIR, combined with microfluidic technology and biomedical knowledge, a series of sensitive new optical sensors, called polarization absorption enhanced biosensors, are designed and put into practical applications. The sensitive sensing of this sensor utilizes the polarization absorption effect of graphene. As the RI n of the upper medium of the graphene is changed, the optical power of the two polarization states changes. Therefore, by simultaneously reading the optical power of the two states of polarization in the reflected light, the RI n of the upper medium of the graphene can be sensed. These sensors have a similar structure to SPR sensors and provide a reliable RI sensing platform for complex fluid systems over a wider dynamic range. Sensitive sensing of the test sample can be achieved by measuring the change in RI produced by the flow of the sample through the microfluidic channel. It can be applied to the biological field to achieve sensitive sensing of a single cell, cell lines, anticancer drugs, specific proteins, and antigenic antibodies.

### 3.1. Theoretical Basis of Polarization Absorption Enhanced Biosensor

Although graphene has an optical absorption of 2.3% as a two-dimensional material of a single atomic thickness, 97% light is wasted and not capable of interacting with graphene [109]. This limits the applications of graphene in many fields. Currently, various methods have been developed to enhance light–graphene coupling, including periodic patterning or coverage of plasma nanostructures on graphene, and double-layer graphene [110,111,112]. However, among these methods, only the first two methods increase the absorption of light coupling by graphene [110,113]. Therefore, the goal of enhancing the light–graphene coupling by an inexpensive and simple method is a great challenge without sacrificing the broadband of graphene.

Ye et al. designed a sandwiched graphene structure. Here, graphene is sandwiched at the interface of medium 1(n_1_) with high RI and medium 2(n_2_) with low RI by a sandwich structure, and the RI of medium 1 and medium 2 is fixed, as shown in Figure 5. The intermediate graphene layer has a complex optical constant of $\hat{n}=n+ik$, where *n* is RI of graphene and *k* is extinction coefficient [114,115]. The thickness *d* is relative to the number of graphene layers (e.g., *d* = 0.34 nm for monolayer graphene). This structure not only has strong optical absorption of TE waves under total internal reflection (TIR) but also increases the broadband absorption of graphene. The energy density distribution of TE and TM waves under the graphene sandwich structure is simulated. It can be seen that when the incident angle is greater than the critical angle, the simulation results show a large energy density. In addition, there is a significant energy density distribution in the graphene layer. The stronger the incident energy, the more the graphene layer absorbs, and the less the reflected energy. It can be said that the reflected energy is directly related to the degree of light absorption of the graphene layer, and higher precision is obtained by comparing reflection with transmission [116]. Therefore, the structure also exhibits different absorption due to the difference in polarized incident waves. For TE and TM waves, there is also a relationship between the incident angle of light on the graphene and the reflectivity. When the angle of incidence rises to a certain angle (about 62°), the reflectance ratio of graphene reaches its maximum and then gradually decreases. The number of layers of graphene also affects the reflectance ratio. When the incident angle is fixed, with the increase of the layers of graphene, the reflectance also increases. In addition, the theoretical and experimental results are basically the same for the reflectance ratio of TM and TE waves. Under the condition of neglecting scattering, TIR prevents transmission from affecting the absorbance of graphene. When TIR prevents transmission, graphene has a greater absorption of TE waves, so the reflectivity of this structure to TE waves is much smaller than that of TM waves. Moreover, the results of spectral measurements of four layers of graphene are similar to the broadband polarization-dependent absorption of single-layer graphene. Besides, the structure enhances the absorption of TE waves in the range from 420 nm to 750 nm. Thereby, the number of layers of graphene can be detected by the polarization absorption characteristics of graphene in a wide spectral range [31]. 

The results show that under TIR, graphene exhibits a large absorption of TE waves, whether it is a single layer, bilayer or few-layer of graphene. Further, the number of layers of graphene can be detected by the reflectance of TM and TE waves. In summary, the light–graphene coupling absorption properties have shown the potential in such application fields. 

### 3.2. Sensors for Single Cell Detection

Cancer cells are a variant of cells. The variant cells are off track, setting their own proliferation rate, and we can detect them once the cells accumulate more than one billion. Cancer cells could evolve to be malignant, which can be harmful to the human body, so the precise detection of cancer cells is crucial [117]. There are many methods to detect single cells, among which the microfluidic cytometry shows obvious advantages, including dynamic cell operation, separation, sorting, reusability, and without contamination. However, these methods also have some restrictions, such as high power laser, high cost, complex structure, and cannot accurately detect the full signal of a single cell [118,119]. In addition, because graphene shows excellent electrical properties, such as its extreme sensitivity to changes in the charge environment, graphene-based FETs have been developed that sensitively sense the flow of single cell [63]. However, it requires measuring current changes on the surface of graphene, and electrical measurements can damage living cells and affect the accuracy of test results [120]. Thus, graphene-based high-sensitivity optical sensors have great potential for the detection of single cells.

Xing et al. designed a graphene-based optical sensor by combining an optical sensor with a microfluidic channel to enable the detection of flow single cell [32]. The graphene-based optical sensor uses high-temperature reduced graphene oxide (h-rGO) and obtains high sensitivity of the sensor by controlling the thickness of graphene. The sensor not only has a high sensitivity of 4.3 × 10^7^ mV/RIU, but also has a higher resolution of RI sensor −1.7 × 10^−8^. This resolution is similar to the resolution (3.8 × 10^−8^) of the capillary-based optofluidic ring resonator designed by Li and Fan [121], slightly lower than the resolution (9 × 10^−9^) of the label-free optical biosensor based on a free-space Young interferometer configuration is presented by Schmitt [122]. In addition, a wide dynamic range can be generated by adjusting the incident angle and incident power. In this study, they used the improved RI sensing model, which inserted graphene layer between high RI medium 1 and low RI medium 2 (Figure 6a), and the mediums 1 and 2 correspond to RI n_1_ and n_2,_ respectively. As the incident angle θ_1_ approaches the critical angle θ_c_, the resolution and sensitivity can be improved by taking advantage of graphene with optimal thickness. Figure 6b shows the reflectance ratio diagram corresponding to angles under different h-rGO thickness. The solid line and the point represent the calculation result and the experimental result, respectively. Calculate the optical constant $\overset{\wedge }{n}=2.6+1.25i$ to confirm the optimal constant $\overset{\wedge }{n}$. In order to achieve high sensitivity of the sensor, h-rGO with a thickness of ~8 nm is the best choice for the sensing layer by calculating the optimal optical constant $\overset{\wedge }{n}$. Generally, single-layer and low-layer graphene grown using CVD exhibit significant advantages, but none of them can be directly applied to a transparent medium but undergo a transfer process. At this time, the problems of surface roughness, peeling, damage, contamination, and folding of graphene are inevitable, and the polarization-dependent absorption effect of graphene will also decrease. At the same time, h-rGO is the best choice compared to other types of graphene.

Resolution and sensitivity are two parameters that are essential for this study and can be calculated by the following formula
(2)$$S=\frac{\frac{d\Delta R}{dn}{|}_{n=n_{2}}\cdot P_{0}}{\alpha }$$
(3)$$L=\frac{V_{n}}{S}$$
where *S* is the sensitivity of RI sensing, *L* is the resolution of RI sensing, Δ*R* is the reflectance difference for TE and TM polarized light, *P*_0_ is the incident power of TE or TM wave, *α* is the response of the balance detector, and *V_n_* is the noise value. In the above formula, some parameters are known quantities. Among them, the power is fixed at 80 μW, the thickness of h-rGO is 8.1 nm, *α* is 0.00361 μW/mV, and *V_n_* is about 10 mV. The h-rGO-based RI sensor detects real-time voltage changes in water and different ultra-low concentrations of sodium chloride solution, as shown in Figure 6c. It can be seen that the difference between the RI of the 0.003% sodium chloride solution ($n=n_{water}+0.000144$) and the RI of the water is about 210 mV. Besides, the illustration shows an enlarged view of the voltage of a 0.01% sodium chloride solution. This shows that the sensor is highly sensitive to flow sensing.

The graphene-based optical single-cell sensor (GSOCS) is shown to be extremely sensitive to single cells by detecting standard polystyrene (PS) microspheres. Figure 6d shows the real-time change in voltage as the mixed 5 and 6 μm PS microspheres roll through the detection window without separation. Their counting and discrimination capabilities are clearly presented with significant differences. The blue and purple line regions represent the discrete voltage signals of the 5 and 6 μm microspheres in the one-time detection diagram, respectively. At high speed of 3 μL/h, the baseline voltage signal stays relatively constant at −5 ± 0.01 V. This result proves that the h-rGO sensor has good sensing repeatability and no surface pollution. Figure 6e shows the entire flow sensing process of single PS microsphere detected by GSOCS. Due to the uniformity of the surface of the microspheres, changes in the voltage signals can be detected at different locations of the single microspheres, and the flow factor of the microfluidic can also be detected. The signal-to-noise ratio (SNR) of a single PS microsphere graph is about 150. Region I-II-III in Figure 6e corresponds to stage I-II-III in Figure 6f. The size and flow rate of a single PS microsphere will affect the time variation (Δt) of the PS microsphere rolling on the detection window. The full width at half maximum (FWHM) of the image can be used to determine this time variation. They have calculated the Δt of a single PS microsphere to be about 28 ms and have shown that the voltage signal can respond quickly in a short time.

In short, they proposed a graphene-based RI sensor that measures the current change of a single cell on the graphene surface. The sensor is capable of a wide range of flow detection with a sensitivity of 4.3 × 10^7^ mV/RIU and a resolution of 1.7 × 10^−8^. They used a GSOCS to detect lymphoma cells from normal lymphocytes and demonstrated that the sensor is highly sensitive. In addition, this high-performance ultra-high resolution and sensitivity sensor for RI measurements can be applied to other areas, such as biopharmaceuticals, medical monitoring, and environmental sensing [32].

### 3.3. Sensors for Anticancer Drugs Detection

Detection of unlabeled living cells has been challenging [123]. The most mature method for unlabeled detection of living cells is using the changes of current or resistance of cell membrane. Although this method shows an extremely fast response, the surface charge of living cells will damage the living cells and affect the response signal of the living cells [30]. At present, a new optical method for detecting RI changes is designed, which could accurately detect the reactions of living cells. In a laboratory, this method has obvious advantages, including no impurity contamination and signal interference, accurate measurement, and dynamic detection. The novel biosensor made by this optical method can quickly and accurately detect the drug reaction in living cells at a high level, and even detect the reaction of cancer cells against cancer drugs [80].

A graphene-based optical biosensor (GOB) consisting of polydimethylsiloxane (PDMS)/microfluidic channel/graphene glass/prism was designed [80], as shown in Figure 7a. The graphene sensing layer, h-rGO is between a low refractive medium and a high refractive medium. The sensor was able to measure an ultra-small RI change (n_c_) of 1.35 × 10^−7^ with an SNR of 5.3 in the experiment, and the corresponding sensitivity was increased to 1.2 × 10^8^ mV/RIU. In addition to hypersensitivity, the sensor can also detect the ultrafast RI changes generated by the weak ultrasound within 260 ns. Due to the ultra-fast response characteristics of GOB, the transient response in living cells can also be accurately detected. Furthermore, the detection depth of the GOB is also very surprising, which is more than 2 μm from the graphene film, and this detecting depth can be used to characterize the height dimension of the entire cancer cell. Therefore, this sensor for living cell sensing has ultra-high sensitivity and high precision. The biosensor is capable of detecting whole-cell responses with the advantage that cell labeling is not needed.

It is extremely important to detect cancer cell responses using high-sensitivity RI sensors over a wide dynamic range. Furthermore, RI changes reflect the detection limits and sensitivity of GOB, which vary greatly within cancer cells. The limit of detection is usually estimated by measuring different sodium chloride concentrations. The ultrasound can accurately evaluate the detection limit, sensitivity, and response time of the GOB. The formula for calculating the detection limit *D* and the sensitivity *S* of the GOB is as follows [124]:(4)S=dU/dn
(5)$$D=N_{noise}/S$$
where *dU* is the change in voltage response and *N_noise_* is the noise signal. The input of the pulse generator not only adjusts the intensity of the ultrasonic pressure but also verifies the detection limit *D* and the sensitivity *S* of the GOB. The ultrasonic generator produces 1 kPa of ultrasound. The relationship between the RI change of the water solution *dn* and the change in the ultrasonic wave pressure dP is *dn* = 1.35 × 10^−10^ dP. As shown in Figure 7b, the GOB detects an ultra-small RI change (1.35 × 10^−7^) produced by 1 kPa of ultrasound, with a corresponding voltage signal of 17 mV and an SNR of 5.3. The noise signal shown in the inset of Figure 7b is approximately 3 mV. In addition, high-pressure ultrasound is used to evaluate the response time of the GOB system, which usually has a longer response time. The response time detected by the GOB is shown in Figure 7c. The noise signal shown in the inset of Figure 7c (on the left) is approximately 2.6 × 10^−8^ RIU. The duration of the entire signal is 2 μs. The illustration in Figure 7c (on the right) shows a direct response time of ~265 ns for the GOB system, which is four orders of magnitude faster than the previous sensor. 

The diameter of cancer cells is larger than normal cells [125]. Two colorectal cancer cell lines, HCT116 and LoVo, are used for measurement. By using a white light interferometer to characterize the height of both cancer cells, it is confirmed that LoVo cells are about 2 μm high, while HCT116 cells are about 1.7 μm. Thus, both cancer cell lines are within the scope of GOB detection. In other words, the detection depth of the GOB and the height of the cancer cells are reasonable, and the sensitivity and accuracy of the living cell sensing are maximized. It is also confirmed here that GOB can detect the response of low concentrations of cancer cells to paclitaxel. In Figure 7d, the RI curves of LoVo cells treated with 6 μΜ paclitaxel solution and LoVo cells treated with no drug are compared. When not treated with drugs, LoVo cells show an almost gradual response within 500 min. When a 6 μM paclitaxel solution is used, LoVo cells show a strong response as the treatment time increase, and gradually approach the saturation trend at 400 min, at which time the saturation response is about 1.9 × 10^−5^ RIU. These results indicate that GOB can accurately measure the response of cancer cells to low concentrations of paclitaxel.

In order to further study the response of cancer cells to paclitaxel, the concentration of paclitaxel solution is used as a variable factor, and 80 nM and 6 μM paclitaxel solutions are used for experiments. The left image of Figure 7e presents a comparison between RI changes in LoVo cells treated with 80 nM and 6 μM paclitaxel solutions. The changes of RI of two kinds in LoVo cells increase rapidly at the start of treatment with paclitaxel solutions at 80 nM and 6 μΜ. This is generally thought to be the reaction of cancer cells to higher concentrations of the drug. It is also found that a larger and faster response is observed in LoVo cells at low concentrations of paclitaxel. When treating with 6 μM paclitaxel solution for 2 min, the response of the cancer cells is approximately linear, and the slope *k* of the response is taken as a statistical parameter to facilitate calculation analysis. The right image of Figure 7e is a partially enlarged image of the left image. As shown in the right figure of Figure 7e, the initial response slope of LoVo cells treated with low concentrations of paclitaxel is much greater than the initial response slope of high concentrations of paclitaxel. This is the first study that has been observed to be contrary to direct. Over time, the response slope of 80 nM paclitaxel-treated LoVo cells decreases faster than the response slope of 6 μM paclitaxel-treated LoVo cells. Furthermore, the response of HCT116 cells to paclitaxel is essentially the same as that of LoVo cells [80]. Figure 7f shows the level of polymerization and all mass of tubulin. Within 10 min after treatment of LoVo cells with 80 nM or 6 μM paclitaxel, the polymerization increases dramatically and gradually stabilizes after about half an hour. However, the amount of microtubule protein increases steadily during the first 4 h and gradually become stable. In addition, comparing the two images of Figure 7f, it could be seen that the 6 μM paclitaxel-induced increase in tubulin mass is significantly higher than that of 80 nM. This indicates that due to the increase in tubulin polymerization and all mass, an increase in RI of early cells is caused. 

In order to better analyze the changes in RI, three factors are summarized here. First, the polymerization and total mass changes of tubulin affected the changes in RI. Secondly, nuclear fragmentation and microtubule purpura affect the changes of RI in some cells of G2/M phase. Finally, after adding paclitaxel, interphase cells take a long time to become apoptotic or necrotic. This also affects the changes in RI. These three factors induce abnormal response curves.

In summary, this biosensor enables ultrasensitive and real-time dynamic detection of the response of unlabeled cancer cells to paclitaxel, and new responses are observed during early drug delivery. Colorectal cancer cells show a stronger and faster response to low-concentration paclitaxel treatment compared to high-concentration paclitaxel treatment. This simple biosensor shows excellent performance for unlabeled living cells and is expected to be widely used as a sensor for cancer treatment, anticancer drug screening and other unlabeled living cell responses for cell-based research [80].

### 3.4. Sensors for Proteins and Antigenic-Antibodies Detection

Since graphene could be easily modified and functionalized for targeted fixation of biological receptor units [126,127,128], graphene shown great application prospects in the biomedical field [129,130]. For example, graphene grown on glass substrates has great potential in biosensor applications. Compared to traditional biosensors, this novel biosensor provides an attractive alternative for real-time detection and monitoring of biomolecular interactions. SPR biosensors are optical sensors that combine graphene to increase sensitivity for detecting, characterizing, and quantifying biomolecular interactions. Under TIR, the coupling of evanescent waves and graphene layers is very different for TE and TM polarized light. Graphene absorbed TE wave more than TM wave. It is very sensitive to changes in the RI of the medium in contact with graphene.

The surface treatment of reduced graphene oxide (RGO) is more likely to enhance the surface function of the SPR sensor relative to the gold film [131]. Furthermore, considering the low cost and mass production of graphene oxide (GO), RGO-based sensors could be easily integrated and manufactured at a low cost [132,133]. RGO-based sensors could be used to detect different concentrations of NaCl solution. With H_2_O as the base liquid, when a 0.011% NaCl solution is introduced, a potential difference between the two liquids is clearly observed. Therefore, the change in RI between H_2_O and 0.011% NaCl solution could be easily seen, indicating that this based RGO sensor is very sensitive to changes in RI. Biomolecular interactions (e.g., antigen–antibody binding, dissociation, and elution) typically induce changes in RI. Since RGO is easy to modify and functionalize, graphene can distinguish between cancer cells and normal cells by detecting biomolecules in the case of TIR [83].

Here, the optical sensor and RGO are combined and functionalized to develop an RGO-based optical biosensor that could be used for binding to specific proteins. The sensor is able to detect biomolecules. A balanced photodetector is capable of measuring changes in the polarization-dependent absorption corresponding to changes in RI induced by antigen–antibody interactions. The sensor responds well to minimal concentrations of immunoglobulin G (IgG) and has the advantages of real-time detection, no labeling, ease of integration, and low cost.

The schematic of an RGO-based optical biosensor is shown in Figure 8. First, the microfluidic channel and graphene glass are combined, and the microfluidic channel and graphene glass are attached to the bottom surface of the K9 glass prism by a suitable index matching liquid. Rabbit IgG (antigen) solution is then passed into the microfluidic channel, and rabbit IgG is better bound to the graphene surface by heating at 35 °C for 1 h. The excess antigen is washed away with phosphate buffer solution (PBS), meanwhile the biorecognition element (antigen) bound to the surface of the graphene to detect specific biomolecules (antibodies) in the solution. The circularly polarized light (532 nm) focused by the objective lens enters the prism and is completely reflected at the interface, and finally separate the reflected light into TE polarized light and TM polarized light. Since graphene absorbs TE waves more than TM waves, balanced photodetectors could be used to record and compare the power difference of the separated light. The inset of Figure 8 gives a schematic representation of the binding of antigen–antibody to the RGO surface. The biosensor can measure the dynamic process of biomolecular interactions in real time. First, PBS is used as a baseline solution to inject the microfluidic channel. Then, the IgG is injected, and the absorption of the TE wave becomes large due to the binding of the antigen–antibody, and the signal shows an upward trend. Subsequently, PBS is injected again to demonstrate that the antigen and antibody no longer reacted. Thereafter, by injecting glycine to elute a large number of antigen molecules bound to the surface of RGO, the real-time signal decreases rapidly. Finally, PBS is injected to raise the signal to baseline, indicating that the antigen IgG molecule completely dissociated from the graphene surface, and the sensor could be used again [82,83].

Sensitivity is an important parameter for biosensors. The sensitivity of the biosensor is investigated by injecting different concentrations of rabbit IgG solution into the microfluidic channel. The minimum concentration of 0.0625 μg/mL rabbit IgG solution can be distinguished with a voltage change of about 0.17 V, and the sensitivity of the sensor is well linear with the rabbit IgG concentration. Specificity is also an important parameter for biosensors. Bovine IgG specificity was also verified in the same conditions for detecting rabbit IgG. Goat anti-rabbit IgG was injected into the microfluidic channel and allowed to bind to the sensor surface, and then bovine IgG was injected until the detected signal was stable. Comparing the results of the test, it could be seen that the voltage change of bovine immunoglobulin M (IgM) is significantly smaller than the voltage change of rabbit IgG, thus indicating that rabbit IgG is not fully bound. Therefore, an RGO sensor modified with an antibody selectively detects an antigen. In other words, different antibodies bind to the biosensor and could match different antigens.

In summary, due to the interaction of antigens with antibodies and the polarization absorption properties of graphene, they designed an RGO-based optical biosensor for detecting properties. Based on the experimental results, the sensor can measure the minimum reaction of the minimum concentration of 0.0625 μg/mL rabbit IgG, which is more sensitive than the previous SPR device. This sensor provides real-time, high sensitivity, and selectivity for the detection of specific proteins without labeling. In addition, considering the use of GO, the sensor can be produced at low cost and on a large scale. These factors should make this sensor a good potential candidate for biosensors.

## 4. Conclusions and Future Works

Graphene is a novel two-dimensional material, due to its unique structure and superior performance. Graphene-based derivatives are used in various fields, including biomedicine and sensing. Among them, the applications in biological imaging, biomolecular detection, cancer detection, drug treatment, etc. is particularly extensive. Although graphene-based electrical sensors are widely used, they can only measure current changes on the graphene surface and may damage living cells, which can affect the test results. Graphene-based optical sensors do not have the above problems, and the detection range is wide, which has potential application prospects in the sensing field. This paper focuses on the optical properties of graphene (such as broadband absorption, saturation absorption, and fluorescence of graphene), the mechanism of light action and the method of enhancing interaction with light, and summarizes the research progress of graphene in the field of sensing, and the structure and characteristics of graphene used in these sensors. Moreover, the interaction between graphene and light under the prism TIR structure is proposed. Under this structure, the interaction between graphene and light is polarization-dependent, and under certain conditions can enhance the absorption of light by graphene. Additionally, under the condition of total internal reflection, the interaction of light with graphene is extremely sensitive to the refractive index of the medium, so the required sensitivity, selectivity, and reusability can be achieved according to graphene in biosensing applications. This paper reviews in detail the latest research on the application of graphene-based optical sensors in the biological field. Graphene-based optical biosensors not only detect whole-cell responses in the presence of unlabeled cells but also show high sensitivity and precision. These simple biosensors have the advantage of large depth detection, high sensitivity and accurate and rapid identification for unlabeled living cells and are expected to be widely used in cell-based research for anticancer drug screening, cancer therapy, and other unlabeled living cells response. These graphene-based optical biosensors have excellent sensitivity and selectivity for detecting single cells, cell lines, anticancer drugs, and specific proteins.

At present, new biosensors and other applications using graphene as a sensing element have been developed, but there is still much work to be done. First, the shape, size, number of layers, electronic band gap structure, purity, and defects of graphene grown in the experiment are all uncertain. However, these properties can affect the conductivity of the sample, interaction with biomolecules, and fluorescence quenching, which, in turn, can affect the performance of graphene-based optical biosensors. Therefore, an efficient method of growing graphene should be explored to obtain high-quality graphene-based nanomaterials [134,135]. Secondly, graphene-based biosensors are used in the limited biomedical field. Many applications are limited to the detection of biomolecules and need to be extended to medical treatment in new research. For example, new devices based on graphene-based nanomaterial design can be used to analyze brain degeneration and neurological disorders, which can prevent in advance and provide protection [136,137]. In addition, researchers should strive to explore the design of new biosensors to adapt to a variety of complex environments [138,139]. For example, graphene-based optical biosensors are put into commercial-scale production and are not limited to laboratory design, which requires optical biosensors to have good reusability and low cost. It is also important to address the possible toxicity and biocompatibility of graphene. Although several functional chemicals have been shown to be biocompatible with graphene, further research is needed on the possible harm of graphene to the human body, so that graphene can be better applied to humans.

In summary, because graphene materials are superior to traditional materials, detailed research on graphene-based materials will open a new direction for research in the field of biosensors. Graphene optical biosensors will also provide critical and breakthrough solutions for the world.

## Figures and Tables

**Figure 1 ijms-20-02461-f001:**
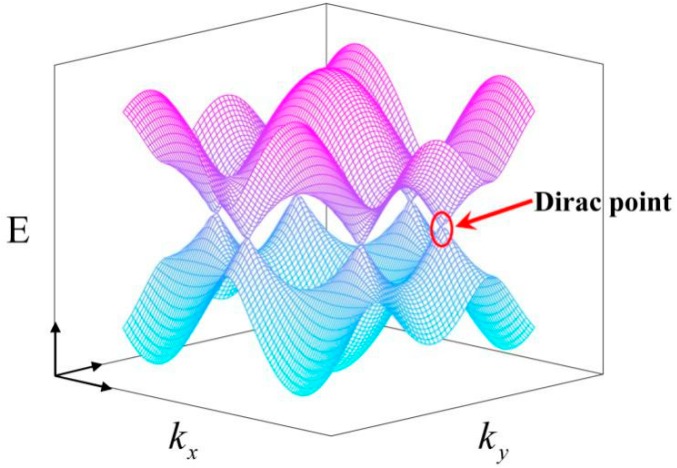
Electron band structure and Dirac point of graphene. Purple represents the n-type doped Fermi surface, and light blue represents the p-type doped Fermi surface.

**Figure 2 ijms-20-02461-f002:**
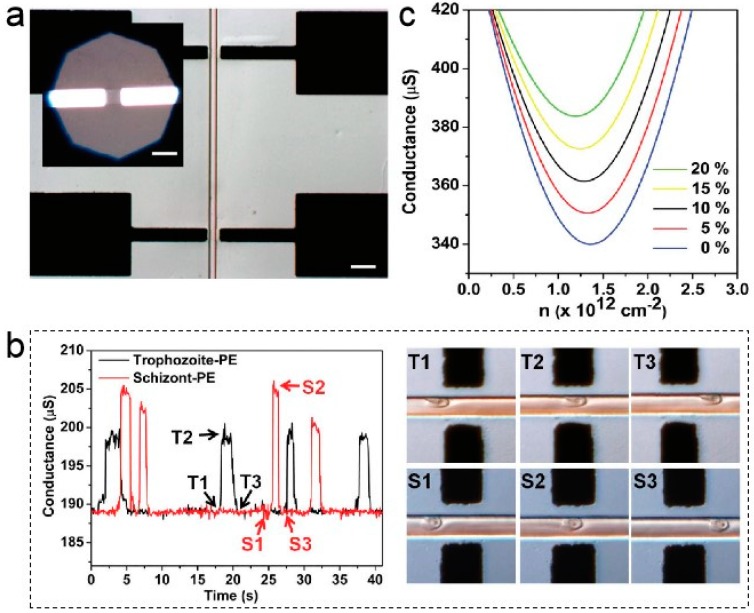
(**a**) Differential interference contrast (DIC) image of independent graphene transistors with SU-8/PDMS microfluidic channel. Inset shows the etched graphene strip between source and drain electrodes. (**b**) Conductance-time plots for (early to mid) trophozoite-PE and schizont-PE measured at Vg = 0.1 V and corresponding DIC images on the right. (**c**) Conductance vs carrier density, n, near CNP for protein-functionalized graphene in solution. Reproduced from [63] with permission of the American Chemical Society.

**Figure 3 ijms-20-02461-f003:**
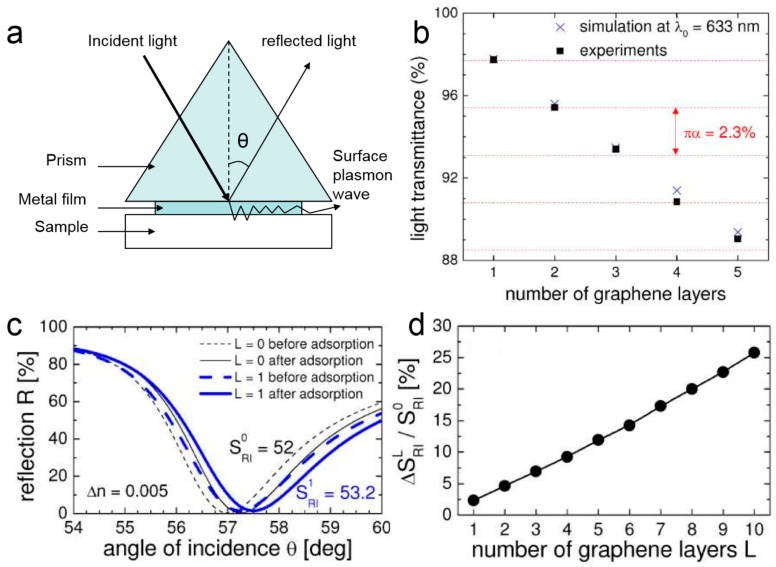
(**a**) The structure of surface plasmon resonance (SPR) sensor and its working principle. (**b**) Simulated transmittance of light at λ_0_ = 633 nm (crosses) and measured transmittance of white light (squares) as a function of the number of graphene layers. (**c**) The SPR curves for the conventional biosensor (L = 0) and the monolayer graphene biosensor (L = 1) for He-Ne laser light. (**d**) The sensitivity enhancement ΔS_RI_^L^/S_RI_^0^ as a function of the number of graphene layers L. Reproduced from [68] with permission of the Optical Society of America.

**Figure 4 ijms-20-02461-f004:**
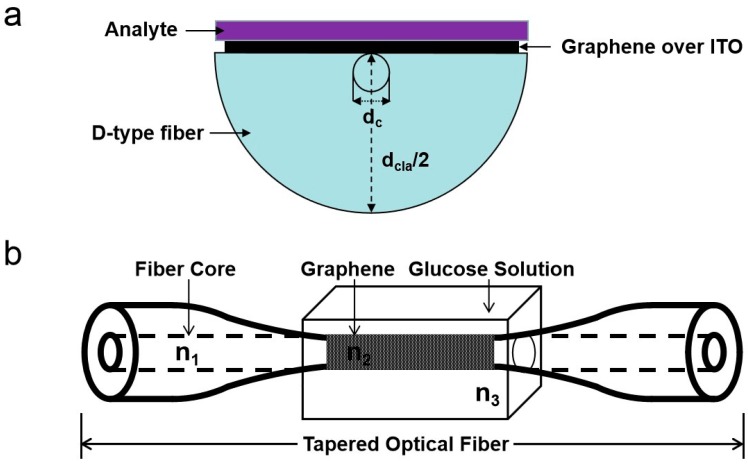
(**a**) Cross-sectional view of the proposed structure. (**b**) The structure of the tapered optical fiber. Graphene is coated on the core of the fiber.

**Figure 5 ijms-20-02461-f005:**
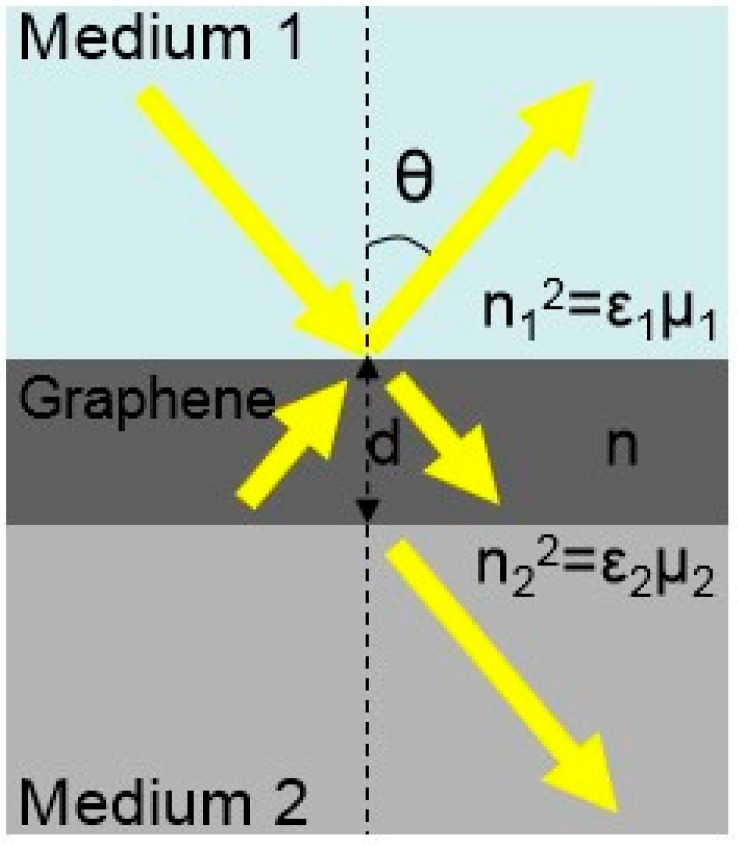
Schematic of a graphene layer sandwiched between two dielectrics.

**Figure 6 ijms-20-02461-f006:**
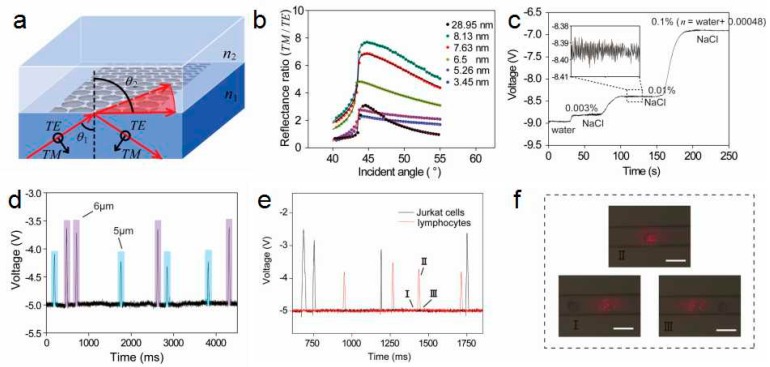
(**a**) Schematic of the enhanced sensitivity and resolution of refractive index (RI) sensing of h-rGO. (**b**) Angle-dependent reflectance ratio (TM/TE) plots of different thicknesses of h-rGO. (**c**) Real-time signal of different ultralow concentrations of NaCl solution. (**d**) Discrete time-dependent changes in voltage that corresponds to PS microspheres as they roll across the h-rGO detection window. The light blue and light purple areas represent the discrete voltage signals of 5 and 6 μm PS microspheres, respectively. (**e**) Discrete time-dependent changes in voltage that corresponds to a single lymphocyte or Jurkat cell as it rolls across the detection window. The black and red lines represent Jurkat cells and lymphocytes, respectively. (**f**) Microscopic images of the h-rGO detection window as lymphocytes roll across it. The scale bar is 15 μm, and the height of the microfluidic channel is approximately 9 μm. Reproduced from [32] with permission of the American Chemical Society.

**Figure 7 ijms-20-02461-f007:**
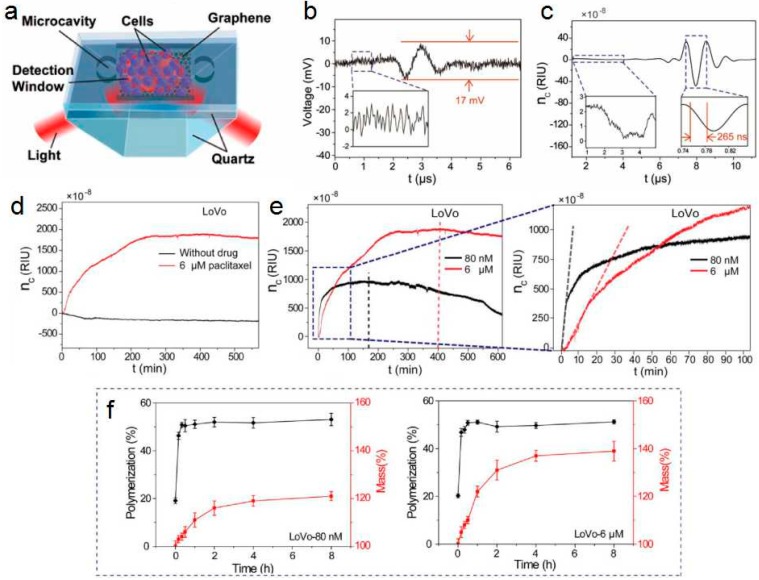
(**a**) Schematic and principle of graphene-based optical biosensor (GOB) sensing. (**b**) The direct detection of a weak RI change generated by an ultrasonic wave. (**c**) The response time of the GOB. (**d**) Detection of cancer cell responses to paclitaxel. (**e**) The time-resolved RI changes in LoVo cells with 80 nM and 6 μM paclitaxel solution. The right is an enlarged image of the left, and the slope analysis of the initial responses. (**f**) Tubulin polymerization and total mass analysis of LoVo cells treated with paclitaxel at different time points. Reproduced from [80] with permission of the Sensors and Actuators B.

**Figure 8 ijms-20-02461-f008:**
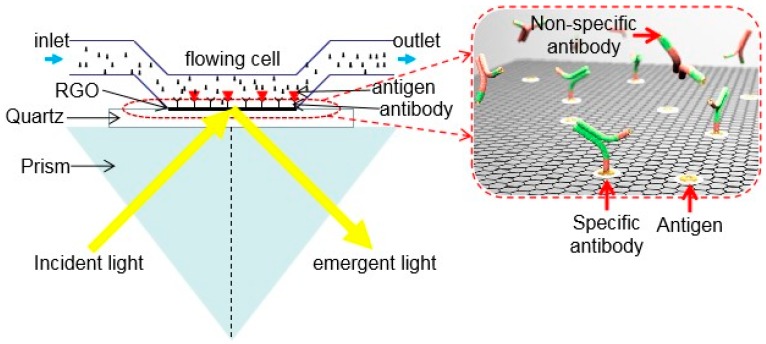
The schematic of reduced graphene oxide (RGO)-based optical sensor.

**Table 1 ijms-20-02461-t001:** The difference between a graphene-based electrical sensor and a graphene-based optical sensor.

	Work Principle	Advantages	Disadvantages	References
Graphene electrical sensor	Since graphene exhibits ambipolar behavior, the p-type or n-type behavior can be tuned effectively by the gate voltage. The principle of sensing is based on changes in drain-source conductivity of the graphene channel upon the binding of the sample to the receptor-functionalized graphene.	Small size, large surface area, fast electron transfer, fast response time, high sensitivity, and reduced surface contamination	Only measure current changes, low spatial resolution, damage samples, affect results	Ang et al. 2011 [63]
Graphene optical sensor	Under total internal reflection, graphene exhibits characteristics of enhanced polarization absorption and broadband absorption. The sensor uses the attenuated total reflection method to detect the refractive index change near the sensor surface.	High spatial resolution, wide and deep detection range, high sensitivity and high precision, accurate and fast detection, unlabeled samples	Since the light absorption rate of single-layer graphene is too low, the area generated by the active photocurrent is too small. Aggregation and precipitation of high concentration samples may affect optical detection.	Wu et al. 2010 [68]

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
