# Peer review of "Graphene Optical Biosensors"

_ijms, 2019, doi:10.3390/ijms20102461_

Round 1

Reviewer 1 Report

This manuscript reports a review on graphene-based optical sensors.  It starts with a description of the unique electronic structure of the carbon-conjugated two-dimensional sp2 system followed by a brief description of its sensor applications in forms of graphene transistors.  In subsequent sections, research progress along with potential and challenges in graphene optical sensors are summarized.

The review is almost complete.  I only have minor comments before the review is accepted for publication:

1.       The authors can provide a table highlighting differences in the electrical and optical sensors based on graphene as it is not clear what discriminates the two in the present form; for instance, the first sentence in page 4 should mention the quantitative difference between the temporal and special resolutions.

2.       It is better explained why section 2 and 3 are separated as polarization absorption enhanced biosensors are also an optical sensor.

3.       Dimension mismatch in equation (1) needs to be checked. 

4.       Figure 1 lacks labels for the three axes.  The color code should also be explained in the caption.

5.       The description “Surface plasmon was a small metal particle” sounds incorrect as plasmon is quantum of plasmonic oscillation. 

6.       Copyright permission would be necessary for the figures of previous literatures.

Author Response

Response to Reviewer 1 Comments

To Reviewers

Dear Sir,

Thank you very much for your comments.

The revisions are made according to your comments.

Point 1: The authors can provide a table highlighting differences in the electrical and optical sensors based on graphene as it is not clear what discriminates the two in the present form; for instance, the first sentence in page 4 should mention the quantitative difference between the temporal and special resolutions.

Response 1: Thanks for reviewer’s kind suggestion and we do agree with the reviewer. We have provided a table in the manuscript highlighting the differences between graphene-based electronic and optical sensors. In addition, We have re-described the first sentence in page 4 as “The developed sensor has an excellent detection limit of paracetamol with a detection limit of 3.2×10-8 M, which has high sensitivity and selectivity for detecting paracetamol in medical applications and commercial fields.”

Point 2: It is better explained why section 2 and 3 are separated as polarization absorption enhanced biosensors are also an optical sensor.

Response 2: We agree with the opinion that polarization absorption enhanced biosensors are one of the optical sensors. The original section 2 of the spatial light sensor has moved to the section 3, which is the theoretical basis of the polarization absorption enhanced biosensors. The section 2 mainly reviews the working principles and applications of graphene-based SPR sensor, while the section 3 reviews the polarization absorption enhanced biosensor. The section 3 is based on the section 2 of the in-depth research, expounding the deeper application of this kind of sensor in biomedicine, which is the main content of Pro. Xing's design of the sensor.

Point 3:Dimension mismatch in equation (1) needs to be checked. 

Response 3: Thank you for your reminding. We have solved this issue and made a modification.

Point 4: Figure 1 lacks labels for the three axes. The color code should also be explained in the caption.

Response : Thanks to the reviewer for the kind reminding. We have modified the figure and re-drawn the Dirac point. In addition, the color code is explained in the caption.

Point 5: The description “Surface plasmon was a small metal particle” sounds incorrect as plasmon is quantum of plasmonic oscillation. 

Response 5: Thank you for your reminding. After careful consideration, we have not described the surface plasmonics as accurate enough and have modified it to be “plasmon is quantum of plasmonic oscillation”.

Point 6: Copyright permission would be necessary for the figures of previous literatures.

Response 6: Thank you for your reminding about the copyright permission on the reusage of previous figures. The figures for obtaining the Copyright permission have been indicated in the manuscript, and the other figures are created by the authors.

Reviewer 2 Report

In “Graphene Optical Biosensors”, the authors present qualities of graphene that can be used for building graphene-based optical biosensors. The paper aims to review recent advances in graphene-based optical sensors and biosensors, highlighting how graphene can bring qualities of ultra-fast detection, high sensitivity and light capture that can be made suitable for single-cell detection, anticancer drug detection and protein and antigen antibody detection.  The review has merit, yet it can be improved to provide the degree of clarity that would do it justice.

With regard to the abstract claim that the article represents a review of recent advances, there are at least 36 publications since 2018 that discuss graphical optical biosensors, yet the authors cite just one publication within this period, their previous study related to cancer cell responses to paclitaxel, so as an overview of the topic suggested by the title, the manuscript is deficient.

From L95 there is a paragraph in the introduction regarding an electrochemical sensor, although  there is little lead-up to explain why the topic should change to electrochemical rather than optical sensors. Then, from L113 a couple of sentences mention criticism for electrochemical sensors, presenting optical sensors as comparatively favourable. Though science is a competitive field, there are advantages and disadvantages for both signal transduction approaches and there is little need to present electrochemical and optical graphene-based sensors as competitors.

The review highlights Graphene SPR sensors and their optical fibre versions, Graphene space light sensors and polarization absorption enhanced biosensors, with emphasis on suitability for applications that include single cell detection, cancer drug response screening and real-time antigen-antibody interaction monitoring.  Conclusions and future works mentions risk factors, including uncertainty regarding graphene quality with very severe limitations for delivering the consistency required for the kinds of application suggested. It even brings into question whether graphene materials are superior to traditional materials. Can the authors point specifically at potential solutions? There are no citations supporting comments in the Conclusion and future works section.

Overall, the field has moved on from mentioning the superlative qualities of graphene to a period when understanding the material properties more profoundly is helping realise how more particular aspects of graphene might be exploited. The concern is that when relying on the limits of performance of the material, the manufacturing quality becomes an increasingly important factor governing measurement outcomes. The authors do well to highlight SPR and particular light capturing qualities that favour coupling graphene to polarised light measurements, but the review could be more carefully structured. Appreciating the multidisciplinary nature of the scientific audience, it would be helpful to have a more didactic approach, with careful subtitle selection. Definition of the acronym SPR first occurred in the introduction rather than under the heading 2.1 Graphene SPR sensors, so a reader unfamiliar with the term would need to hunt for its definition.

The authors have important insights concerning qualities of graphene favourable for its use within optical biosensor frameworks, but there needs to be either a more extensive description of recent complementing literature or a rephrasing of the main title to remain more consistent with the content. 

Additional corrections:

L21 materials > material

L82 sensitive > sensitively

L431 grapheme > graphene

L436 can > could

L480 um > µM

L440 The biosensor is capable of detecting whole cell responses without the advantage of labeling. > I believe the authors mean that the biosensor is capable of detecting whole cell responses with the advantage that cell labelling is not needed.

L461  “The diameter of cancer cells is larger than normal cells”…Although the statement is well supported by an excellent review from Prof Sherr, our perspectives regarding cancer cells have moved on since 1996 and it would always be regarded as oversimplification to assume that all cancer cells can be distinguished purely on the basis of cell diameter.

Author Response

Response to Reviewer 2 Comments

To Reviewers

Dear Sir,

Thank you very much for your comments.

The revisions are made according to your comments.

Point 1: With regard to the abstract claim that the article represents a review of recent advances, there are at least 36 publications since 2018 that discuss graphical optical biosensors, yet the authors cite just one publication within this period, their previous study related to cancer cell responses to paclitaxel, so as an overview of the topic suggested by the title, the manuscript is deficient.

Response 1: Thank you for your kind reminding. Indeed, graphene optical biosensor is a hot topic in recent years, and it is a giant work to review all the graphene optical biosensors. This is an invited review manuscript concerning about Prof. Xing's recent research, it is mainly a review of Prof. Xing's recent biosensing research based on graphene optical sensors. As a result, research on our own is provided and the recently sensors in this field have not been described in detail recently. Besides, we have revised the title of the section.

Point 2: From L95 there is a paragraph in the introduction regarding an electrochemical sensor, although there is little lead-up to explain why the topic should change to electrochemical rather than optical sensors. Then, from L113 a couple of sentences mention criticism for electrochemical sensors, presenting optical sensors as comparatively favourable. Though science is a competitive field, there are advantages and disadvantages for both signal transduction approaches and there is little need to present electrochemical and optical graphene-based sensors as competitors.

Response 2: Thank you for your kind reminding. We are sorry for the misunderstanding. We don’t intend to criticize graphene-based electrical sensors. We just want to use the disadvantages of graphene-based electrical sensors to bring out optical sensors. We have supplemented the advantages of electrical sensors in the manuscript and reorganized them.

Point 3: The review highlights Graphene SPR sensors and their optical fibre versions, Graphene space light sensors and polarization absorption enhanced biosensors, with emphasis on suitability for applications that include single cell detection, cancer drug response screening and real-time antigen-antibody interaction monitoring.  Conclusions and future works mentions risk factors, including uncertainty regarding graphene quality with very severe limitations for delivering the consistency required for the kinds of application suggested. It even brings into question whether graphene materials are superior to traditional materials. Can the authors point specifically at potential solutions? There are no citations supporting comments in the Conclusion and future works section.

Response 3: Thanks to the reviewer for pointing out this issue. Graphene materials have many advantages that are not available in conventional materials and are widely used. At present, many teams have also conducted research on the quality of graphene. G. Z. Wang team grew a large area of high quality graphene on different types of copper foil preannealed under positive pressure H2 ambience. The prepared graphene showed good electrical conductivity, transmittance, and uniformity. The sheet resistance values of the grown monolayer graphene film were all about 500Ω,and the transmittance was as high as 97.24%. H. M. Cheng and W. C. Ren team can quickly obtain a continuous uniform single-layer graphene film on liquid Cu in 3 minutes. Compared to the original film grown on solid Cu foil, the obtained film shows greater grain size, higher quality, better optical and electrical properties, and better performance in OLED applications. Besides, we have added citations to support the comments in the Conclusion and future work section.

Point 4: Overall, the field has moved on from mentioning the superlative qualities of graphene to a period when understanding the material properties more profoundly is helping realise how more particular aspects of graphene might be exploited. The concern is that when relying on the limits of performance of the material, the manufacturing quality becomes an increasingly important factor governing measurement outcomes. The authors do well to highlight SPR and particular light capturing qualities that favour coupling graphene to polarised light measurements, but the review could be more carefully structured. Appreciating the multidisciplinary nature of the scientific audience, it would be helpful to have a more didactic approach, with careful subtitle selection. Definition of the acronym SPR first occurred in the introduction rather than under the heading 2.1 Graphene SPR sensors, so a reader unfamiliar with the term would need to hunt for its definition.

Response 4: We do agree with the reviewer's view. Manufacturing quality is an increasingly important factor in measuring measurement results. This review focuses on polarization-absorption enhanced biosensors that do not describe much of SPR and specific light-capturing properties. In addition, we have modified the abbreviation SPR in the heading to “surface plasmon resonance”.

Point 5: The authors have important insights concerning qualities of graphene favourable for its use within optical biosensor frameworks, but there needs to be either a more extensive description of recent complementing literature or a rephrasing of the main title to remain more consistent with the content.

Response 5: Thank you for your comments. We have re-described the main title to remain more consistent with the content.

Point 6: Additional corrections:

L21 materials > material

L82 sensitive > sensitively

L431 grapheme > graphene

L436 can > could

L480 um > µM

Response 6: Thank you for the kind reminding. We have fixed these mistakes and completed the modification to these errors in the manuscript.

Point 7: L440 The biosensor is capable of detecting whole cell responses without the advantage of labeling. > I believe the authors mean that the biosensor is capable of detecting whole cell responses with the advantage that cell labelling is not needed.

Response 7: We agree with said the view. The meaning of this sentence is that the biosensor is capable of detecting whole cell responses with the advantage that cell labelling is not needed, and we have made corresponding correction in the manuscript.

Point 8: L461  “The diameter of cancer cells is larger than normal cells”…Although the statement is well supported by an excellent review from Prof Sherr, our perspectives regarding cancer cells have moved on since 1996 and it would always be regarded as oversimplification to assume that all cancer cells can be distinguished purely on the basis of cell diameter.

Response 8: Your view is reasonable. We agree that not all cancer cells can be distinguished purely on the basis of cell diameter. Referring to the diameter of cancer cells, it is mainly proved that the detection depth of the sensor and the height of the cancer cells are reasonable. The end is to show that the sensor can accurately measure the response of cancer cells to low concentrations of paclitaxel.

Reviewer 3 Report

In this paper titled “Graphene Optical Biosensor”, the authors reviewing the optical properties of graphene and the implementation of both graphene and graphene derivates as a biosensor. These are the comments raised by the reviewer to improve the quality of current work.

1. Based on this reviewer’s opinion, Authors should include the permission of each figure from the original journal. Please edit the caption of the figures by referring to other review articles.

2. Graphene-based optical sensor part: The authors are encouraged to change the format of the introduction section. Please refer to other articles.

3. The affiliation of the researchers is not necessary to be cited in this manuscript. Please delete it.

4. Following abbreviation are recommended to be explained

- CVD (Page 7, Line 233)

5. Following sentences are recommended to be corrected.

- “Surface plasmon was a small metal particle produced by photon-excited electron resonance,” (Page 5, Line 145) should be corrected as “Surface plasmon is a small metal particle produced by photon-excited electron resonance,”

- “grapheme sensing layer, h-rGO is between a low refractive medium and a high refractive medium.” (Page 12, Line 431): Grapheme should be corrected as 'graphene'.

Author Response

Response to Reviewer 3 Comments

To Reviewers

Dear Sir,

Thank you very much for your comments.

The revisions are made according to your comments.

Point 1: Based on this reviewer’s opinion, Authors should include the permission of each figure from the original journal. Please edit the caption of the figures by referring to other review articles.

Response 1: Thank you for the kind reminding. All figures have been copyrighted except the author's own figures. Besides, I have edited the captions of the figures by referring to other review articles.

Point 2: Graphene-based optical sensor part: The authors are encouraged to change the format of the introduction section. Please refer to other articles.

Response 2: Thank you for the suggestion. We have modified the graphene-based optical sensor section.

Point 3: The affiliation of the researchers is not necessary to be cited in this manuscript. Please delete it.

Response 3: Thank you for your reminding. We have deleted the affiliation of the researchers in the manuscript.

Point 4: Following abbreviation are recommended to be explained CVD (Page 7, Line 233)

Response 4: Thanks you for your suggestion. CVD is an abbreviation for Chemical Vapor Deposition and has been revised in the manuscript.

Point 5: Following sentences are recommended to be corrected. “Surface plasmon was a small metal particle produced by photon-excited electron resonance,” (Page 5, Line 145) should be corrected as “Surface plasmon is a small metal particle produced by photon-excited electron resonance,”

“grapheme sensing layer, h-rGO is between a low refractive medium and a high refractive medium.” (Page 12, Line 431): Grapheme should be corrected as 'graphene'.

Response 5: We are sorry for the mistakes. Thanks to the reviewer's correction. “Surface plasmon was a small metal particle produced by photon-excited electron resonance,” has been corrected as “Surface plasmon is quantum of plasmonic oscillation generated by the interaction of freely vibrating electrons and photons on a metal surface,”. This description will be more accurate. In addition, Grapheme has been corrected as 'graphene'.

This manuscript is a resubmission of an earlier submission. The following is a list of the peer review reports and author responses from that submission.

Round 1

Reviewer 1 Report

This review is not well organized. The authors often mention the Figures  in References.

It is not easy to differentiate the figure numbers in this review.

In general, fig. number in references should not be mentioned whether it is original paper or review.

This reviews explain limited works(Ref. 36, 87, and 90) of biosensors. Other reference should be mentioned more and compared. Some previous works mentioned in the text are not shown as reference.

Reviewer 2 Report

The respected authors present an overview on optical biosensors based on graphene. First, with a comprehensive and well-written overview over the field, introducing the material graphene itself, to then present a few electrical, electro-chemical and then optical sensing methods based on graphene.

Unfortunately, after a very nice introduction, and despite the fact that the manuscript covers multiple studies from various fields, the paper fails to thoroughly review the field, to draw conclusions, to connect the dots, no explanation or attempt to link the publications. It's more bits and pieces of different publications with no "glue" in between. These pieces seem tobe mostly from the Materials & Methods as well as the Results section of said papers, making it extremely confusing to read. E.g., there is often too much detail provided for certain reviewed papers (symptomatic e.g. between lines 463 - 484) for being a review paper. Same holds true e.g. for Figure 7, which is already very specific and difficult to digest for a normal scientific paper.

Additionally, despite the 145 references, the manuscript mostly focuses on 7 or 8 specific papers only, which does not provide a broad overview over the field. Also, unfortunately, Western literature seems to be treated negligently. Often the experimental setup and how the measurement is performed is not described adequately. One of many examples is line 443: “… produced by 1 kPa ultrasound”. It remains completely unclear, where the ultrasound comes from and what it has to do with refractive index changes and how this is transduced into millivolts.

Moreover, the manuscript often references very specific pictures in a reference. Either the picture should be redrawn if crucial for the review paper, or the concept should be well explained. Otherwise this should be omitted. In general, figures for a review paper should be redrawn or permission should be requested from the corresponding journals.

Other “Don’ts” are: several times in the manuscript (incl. the Abstract), the authors predict future events like “These new high-performance [..] optical sensors will be able to detect surface structural changes [..].” (line 14) or “In summary, the light-graphene couplingabsorption properties will have great potential in many applications in the field of optics”. I’m afraid that speculations don’t belong in a review paper.

Additionally and unfortunately, the manuscript has confusing, imprecise phrases, potentially due to the language barrier, such as “Cancer cells are derived from “rebellious” normal cells” or “Cancer cells are higher than normal cells”. Higher than what? Moreover, many abbreviations are not explained, such as LoVo.

Some claims are most certainly not correct (line 347): highest resolution of RI sensors. WGM sensors and interferometric sensors report 10-9 since years. See e.g. H. Li and X. Fan, , Appl. Phys. Lett., 2010, 97, 011105 or Schmitt, K., Schirmer, B., Hoffmann, C., Brandenburg, A., Meyrueis, P., 2007. Biosens. Bioelectron. 22, 2591–2597.

Last but not least: Summary (very important for a review paper is completely missing, and conclusion and outlook extremely short.